# Pharmaceuticals and Microplastics in Aquatic Environments: A Comprehensive Review of Pathways and Distribution, Toxicological and Ecological Effects

**DOI:** 10.3390/ijerph22050799

**Published:** 2025-05-20

**Authors:** Haithem Aib, Md. Sohel Parvez, Herta Mária Czédli

**Affiliations:** 1Pál Juhász-Nagy Doctoral School of Biology and Environmental Sciences, University of Debrecen, 4032 Debrecen, Hungary; sohel@nstu.edu.bd; 2Department of Hydrobiology, University of Debrecen, 4032 Debrecen, Hungary; 3Department of Oceanography, Noakhali Science and Technology University, Noakhali 3814, Bangladesh; 4Department of Civil Engineering, University of Debrecen, 4028 Debrecen, Hungary; herta.czedli@eng.unideb.hu

**Keywords:** aquatic environment, ecological effects, ecotoxicology, fish, growth, microplastics, pharmaceuticals, physiological effects, reproduction

## Abstract

Pharmaceuticals and microplastics are persistent emerging contaminants that pose significant risks to aquatic ecosystems and ecological health. Although extensively reviewed individually, a comprehensive, integrated assessment of their environmental pathways, bioaccumulation dynamics, and toxicological impacts remains limited. This review synthesizes current research on the environmental fate and impact of pharmaceuticals and microplastics, emphasizing their combined influence on aquatic organisms and ecosystems. This review provides a thorough and comprehensive examination of their predominant pathways, sources, and distribution, highlighting wastewater disposal, agricultural runoff, and atmospheric deposition. Studies indicate that pharmaceuticals, such as antibiotics and painkillers, are detected in concentrations ranging from ng/L to μg/L in surface waters, while MPs are found in densities up to 106 particles/m^3^ in some marine and freshwater systems. The toxicological effects of these pollutants on aquatic organisms, particularly fish, are discussed, with emphasis on bioaccumulation and biomagnification in the food chain, physiological effects including effects on growth, reproduction, immune system performance, and behavioral changes. The ecological consequences, including disruptions to trophic dynamics and ecosystem stability, are also addressed. Although valuable efforts, mitigation and remediation strategies remain inadequate, and further research is needed because they do not capture the scale and complexity of these hazards. This review highlights the urgent need to advance treatment technologies, establish comprehensive regulatory frameworks, and organize intensive research on long-term ecological impacts to address the environmental threats posed by pharmaceuticals and microplastics.

## 1. Introduction

Globally, there has been an increase in concern over water quality [1]. The existence of emerging contaminants (ECs) in the environment has drawn more attention and has been the subject of extensive recent research [2]. The Federal Water Pollution Act of 1948, which was later renamed the Clean Water Act, required the monitoring and control of dangerous pollutants in freshwater throughout the United States (33 U.S. Code §1251, 1972), although these initiatives have not prevented nutrient pollution, and in recent decades, new contaminants of emerging concern (CECs) have been acknowledged [3,4]. Pharmaceutical and microplastic contamination of aquatic environments is a serious public health and ecological concern [5,6,7]. Microplastics, defined as plastic particles smaller than 5 mm, are significant environmental contaminants in aquatic ecosystems. They can be categorized into primary microplastics, which are intentionally produced for commercial use, and secondary microplastics, formed from the degradation of larger plastic debris. Commonly composed of polymers like polyethylene (PE), polypropylene (PP), and polystyrene (PS), microplastics often contain chemical additives that can leach into the environment, intensifying their ecological impact. Due to their small size, they can be ingested by aquatic organisms, leading to bioaccumulation and potential trophic transfer, which raises concerns about their effects on biodiversity and human health [8,9].

Environmental persistent pharmaceutical pollutants (EPPPs) such as analgesics, hormones, and antibiotics are a matter of great concern when ingested by non-target organisms, and their existence tends to slowly degrade and spread in the environment [10]. These residues of pharmaceuticals have been noticed in surface water and groundwater across the globe [11,12,13]. Conventional wastewater treatment facilities are not made to remove pharmaceuticals from wastewater, and high concentrations of pharmaceutical residues have been discovered downstream of pharmaceutical manufacturing facilities [14]. Furthermore, veterinary pharmaceutical residues from agriculture and aquaculture can enter water bodies without any treatment [15].

Figure 1 shows the numerous pathways from pharmaceutical production to ecosystems, emphasizing how contaminants from industrial sources, wastewater discharge, and runoff from agricultural practices can enter water bodies and accumulate in the environment. These processes also play a part in the rising concentrations of drug residues in aquatic environments. Similarly, microplastics are common environmental contaminants that are released directly from personal care products and industrial processes, as well as from the fragmentation of larger plastic objects, synthetic fiber shedding, and tire abrasions [16]. Because of their malleability and ease of use, plastic was first widely used after World War II, and its production has only increased since then. From 2 million tons in 1950 to 380 million tons in 2015, the world’s production of resin and fiber increased [17]. They are present in voluminous different soils, rivers, oceans, and the atmosphere. Although microplastics have some potential advantages, such as their application in industry and medicine, the disadvantages greatly exceed the advantages [16]. The mechanisms of environmental degradation and the pathways of microplastic pollution are shown in Figure 2. It depicts the accumulation through numerous processes, such as fragmentation of larger plastic items and shedding from synthetic fibers. Similar to pharmaceuticals, which present serious threats and risks to aquatic ecosystems.

The presence of these pollutants poses a significant threat to aquatic ecosystems worldwide. Veterinary antibiotics in European aquatic environments are leading to long-term ecological risks [18,19,20]. The increasing presence of these compounds in river ecosystems poses a significant threat with detrimental effects on humans, animals, and aquatic life. Several studies have critically examined these impacts, along with various remediation strategies [21]. Similarly, microplastics present unique environmental challenges. During polymer manufacturing, chemical additives such as plasticizers, heat stabilizers, antioxidants, and colorants are commonly incorporated to enhance the performance of final products, further complicating their environmental impact. When these compounds are in the environment, they cause chronic risks [22] and significant danger to marine ecosystems [23,24]. Microplastics can accumulate in the human body via different routes comprising inhalation, ingestion, and dermal exposure [24]. Consequently, this accumulation causes an evident negative impact on our health. Numerous studies indicate that microplastics may cause genotoxicity in human cells [25], cellular impairment [26], and inflammation [27]. Pharmaceuticals and microplastics have numerous adverse effects on aquatic organisms, particularly fish. Several scientists and researchers have highlighted the effects in detail, including the influence of painkillers and hormones on the endocrine system and reproductive cycle of fish [28], long-term toxicity, hormonal imbalances, and alterations in fish behavior [29], changes in behavior, liver damage and oxidative stress under the effect of combined drugs and microplastics exposure to fish [30], which frequently become exposed by sediment and water. Given the numerous instances of their harmful effects on both human health and the environment, it is now widely acknowledged that contaminants pose a serious cause for concern regardless of the economic status of a nation [2]. EC management is incredibly difficult. From the standpoint of policy and regulation, it necessitates modifications to present procedures, such as the creation of ambitious but workable policies to address pollutants that have not yet received sufficient research and contaminants that continue to raise concerns [2].

In addition to addressing technological gaps, fostering international collaboration is essential. This involves not only sharing best practices but also investing in research that focuses on the cumulative and long-term effects of these pollutants on aquatic ecosystems. Earlier reviews on this topic have primarily focused on isolated aspects, the influences of microplastics on toxicity, and the transgenerational effects of pharmaceuticals [14,15,16]. This review seeks to address the existing gap in research by comprehensively examining the pathways, bioaccumulation, and toxicological effects of pharmaceuticals and microplastics, focusing on their direct and indirect implications for public health. It highlights the interplay between these pollutants and identifies critical inadequacies in current mitigation strategies. Figure 3 illustrates the global presence of pharmaceutical pollutants and it emphasizes how urgently coordinated oversight and regulatory action are required. By integrating scientific evidence with policy recommendations, this study emphasizes the urgent need for global intervention and transdisciplinary approaches to manage their combined environmental and health impacts.

Pharmaceuticals and microplastics are widespread aquatic micropollutants, and WWTPs are often ineffective at fully removing them. As illustrated in Table 1, various classes of pharmaceuticals including antibiotics, painkillers, and hormones—have been detected in WWTP influents and effluents at concentrations ranging from a few nanograms per liter to several micrograms per liter, depending on the compound and the wastewater source. Similarly, Table 2 summarizes the main types of microplastics found in aquatic systems, such as fragments, fibers, and beads, highlighting their sources and typical particle size ranges. Although the tables have been organized for clarity, the content was directly adapted from previously published sources to maintain accuracy and data integrity.

## 2. Sources and Distribution of Pharmaceuticals and Microplastics

A systematic literature search was carried out using databases like Web of Science, Scopus, PubMed, Google Scholar, and ScienceDirect to ensure a broad and in-depth understanding of the subject. Using terms like “pharmaceuticals in aquatic environments”, “microplastics pollution”, “toxicology”, “bioaccumulation”, and “ecotoxicology”, the search concentrated on pathways, toxicological effects, and mitigation techniques. Articles that were unrelated to aquatic ecosystems or lacked thorough data were excluded, leaving only those published between 2000 and 2024. A rigorous selection process, guided by the PRISMA framework, was employed to ensure that only relevant and high-quality studies were included (see Figure 1). Figure 2 was adapted from a study published in *Science of The Total Environment*.

### 2.1. Wastewater Discharge and WWTP Effluents

#### 2.1.1. Pharmaceuticals

A variety of domestic, agricultural, and industrial sources continuously discharge wastewater, which can leach pharmaceuticals into freshwater ecosystems and lead to drug contamination [38,39]. This contamination is primarily due to human and animal waste [40]. When taking medication, the active ingredients are only partially metabolized; the rest is excreted in the urine or feces [39,41,42]. Toilets and drains serve as primary entry points for these drug residues into the sewer system [43,44]. Veterinary medicines also contaminate aqueous systems, including wastewater [5]. Drugs such as hormones, antibiotics, and other substances commonly administered to livestock and pets can be excreted and enter the sewer system via manure or direct discharge [45,46]. Combined with the improper disposal of unused prescription medications, chemicals from pharmaceutical manufacturing facilities can also end up in wastewater [5,6]. If improperly managed or if treatment processes do not function properly, these facilities could release pharmaceutical compounds directly into wastewater. Pharmaceuticals ultimately end up in wastewater treatment plants (WWTPs) via sewer pipes after being discharged into the sewage system [47,48]. Wastewater is treated in WWTPs using various techniques to remove contaminants before the treated water is discharged into surface waters [49], but not all pharmaceutical compounds may be completely eliminated by conventional wastewater treatment methods [50]. Conventional wastewater treatment methods typically involve a series of procedures designed in three main stages—preliminary, primary, and secondary treatment—often followed by tertiary or advanced treatment and sludge management. Each stage plays a significant role in lessening or removing pollutants from the wastewater as much as possible before being discharged back into the natural systems or reused.

Certain pharmaceuticals are not effectively degraded in standard treatment procedures and are physiologically effective even in low concentrations [29]. Consequently, drug residues can be found in wastewater effluents that have been treated and discharged into surface waters. Different classes of pharmaceuticals could be found in the wastewater such as contrast materials (iohexol, iotalamic acid, iopamidol, iopromide, iomeprol, amidotrizoic acid, diatrizoate), antidepressants (fluoxetin), anti-inflammatories and analgesics (4-aminoantipyrine, antipyrin, codein, diclofenac, ibuprofen, indomethacine, ketoprofen, ketorolac, naproxen), psycho-stimulants (caffeine, paraxanthin), and antibiotics (clarithromycin, ciprofloxacin, doxycyclin, erythromycin, methronidazole, norfloxacin, ofloxacin, roxithromycin, sulfamethoxazole, sulfapyridin, tetracyclin, trim ethoprim) can be found in these wastewaters [51,52,53,54,55].

WWTPs play a crucial role in purifying wastewater before discharging it into surface waters [48,49,56]. However, despite these efforts, certain pharmaceutical substances can bypass conventional treatment methods and enter the environment via WWTP effluent [50]. Human excretion is the main route for pharmaceuticals to reach WWTPs via domestic wastewater [57,58]. These compounds enter the sewage system through drains, sinks, and toilets in residential areas, clinics, and other facilities [43,44]. In addition, veterinary drugs from pet and livestock farms can also contribute to drug contamination in WWTPs [5]. Farm animals are often given antibiotics, hormones, and other drugs, which are excreted and can end up in sewers. Pharmaceutical manufacturing facilities represent another important source of pharmaceuticals in WWTPs, as these compounds can enter sewers directly during the manufacturing process [59] or due to improper disposal practices [50]. Upon entering WWTPs, pharmaceuticals undergo various treatment processes, including chemical, biological, and physical steps [60,61,62], to remove contaminants from wastewater. While these treatments effectively remove many pollutants, some pharmaceutical compounds persist and may remain in the treated wastewater [48,63]. Factors such as the chemical composition of drugs [61], the efficiency of treatment processes, and the design of WWTPs can influence the persistence of these compounds. Consequently, wastewater discharged from WWTPs into surface waters contains [48,57,63] residues of pharmaceuticals, including hormones, antidepressants, analgesics, and antibiotics.

#### 2.1.2. Microplastics

Wastewater discharge significantly contributes to plastic pollution in aquatic environments [64]. Microplastics can enter aquatic ecosystems through the direct discharge of untreated and insufficiently treated wastewater [65]. This includes the microplastics carried by domestic wastewater from washing machines, outlets, and showers, as well as basin waters are one of the main sources of MPs in water bodies. Laundry wastewater, in particular, releases significant amounts of microfibers from synthetic textiles like polyester. A single wash may release up to 1900 fibers [66,67]. Additionally, personal care products containing microbeads and synthetic microplastics contribute to this pollution load [68]. These MPs get into the sewage and finally pile up in water bodies if not treated.

These fibers are carried away by wastewater, entering sewage systems, and, without proper treatment, accumulating in water bodies, where they can persist for years due to their resistance to degradation [35,69].

Microbeads, which are small plastic particles often included in cosmetic and hygiene products, are washed off during daily routines and can contribute significantly to microplastic load in wastewater systems [70]. The inadequacy of current wastewater treatment technologies in removing microplastics has been widely recognized. Enhanced filtration methods, such as membrane bioreactors and advanced physical or chemical treatments, have shown potential in reducing microplastic contamination in effluent [71,72].

Wastewater treatment plants are not always capable of trapping MPs from wastewater [73,74]. It is mainly because of their tiny sizes that MPs allow them to get past the filtering devices used in traditional treatment systems [75]. As a result, MPs remain on the effluents even after being treated. These effluents could be a large contributor of MPs to the environment [76]. However, plastic particles could degrade into smaller sizes due to the chemicals used and the mechanical agitation [77]. These MPs can amass and endanger aquatic systems following their release into the water bodies. While wastewater treatment plants are a critical barrier between anthropogenic pollutants and natural aquatic ecosystems, they are not fully effective in removing microplastics from the wastewater stream [78,79,80].

This inefficiency is primarily attributed to the microscopic size, low density, and buoyant nature of microplastics, which allow them to evade conventional physical and mechanical filtration processes employed in standard wastewater treatment systems [59,81]. As a result, treated effluents discharged into receiving water bodies may still contain substantial concentrations of microplastics, rendering wastewater treatment plants both a sink and a secondary source of microplastic pollution in aquatic environments [78,80]. While the primary and secondary treatment processes (e.g., sedimentation and activated sludge) can remove much of the larger plastic debris, smaller particles, particularly those of nanometric size (<1 mm), often circumvent these systems and are released into the final wastewater [81,82]. The studies show that the removal efficiency ranges from 72 to 99 percent, depending on the design of the plant and the type of treatment used, but that still leaves millions of particles per cubic meter of treated water [81,82].

Furthermore, microplastic particles can be further degraded and fragmented during the sewage treatment process as a result of exposure to chemical oxidants (e.g., chlorine, ozone) and mechanical agitation (e.g., aeration, tank turbulence) [79,81]. This will not only increase particle numbers but also change their physical properties, potentially making them more bioavailable and more reactive [78,81]. Smaller MP and nanoparticles are of particular concern as they are more easily absorbed by aquatic organisms and may pass through bio-molecular membranes, thus increasing the risk of bioaccumulation and trophic transmission [59,82]. Once released, microplastics may settle in sediments, be transported across long distances, infiltrate drinking water sources, be carried by water currents over long distances, and even enter drinking water sources [79,83].

High-tech treatment technologies such as membrane bioreactors, rapid sand filtration, coagulation, and advanced oxidation processes have been shown to have a higher rate of MP removal but have not been widely deployed due to high operating costs and energy requirements [59,82]. Therefore, there is an urgent need to upgrade the current WWTP infrastructure and develop standardized monitoring protocols for the assessment of the effectiveness of microplastic removal [78,81].

In order to reduce the first-degree input of MP into wastewater streams, policy action, public awareness, and innovation at the product level (e.g., microfiber filters on washing machines) are also essential [83,84].

### 2.2. Agricultural Runoff

#### 2.2.1. Pharmaceuticals

Agricultural runoff can serve as a significant pathway for introducing pharmaceuticals, such as hormones and antibiotics, into freshwater ecosystems [56,57,58]. These substances can be sprayed on livestock or crops [5], with the potential to infiltrate groundwater and soil before ultimately emerging as surface waters [46,51,59]. Pharmaceuticals from agricultural runoff primarily enter freshwater ecosystems through the use of crop protection chemicals and veterinary pharmaceuticals [60]. Veterinary antibiotics are increasingly used in various regions to protect animal health and support treatment while also enhancing feed efficiency in aquatic animals, poultry, pets, livestock, silkworms, bees, and other species [61,62]. Animal excretion of these pharmaceuticals can be released into the environment through urine and manure. Apart from veterinary pharmaceuticals, agricultural runoff can also introduce pharmaceuticals like herbicides, fungicides, and insecticides that are used in crop production into freshwater ecosystems [63,73]. When crops are irrigated or exposed to rainfall, the pharmaceuticals applied for pest and disease control can wash off, ultimately reaching surface waters [51]. In freshwater ecosystems, pharmaceutical pollution can also result from the use of biosolids, nutrient-rich organic materials obtained from sewage sludge, as fertilizer in agriculture [39,40,74,75]. Human excretion of pharmaceutical residues may be present in biosolids [76,77]. These pharmaceutical residues can infiltrate the groundwater and soil, eventually reaching surface waters through runoff [78,79]. The widespread distribution of these compounds occurs when agricultural practices introduce them into the soil, from which runoff and leaching processes transport them to nearby surface waters. The likelihood of contamination is influenced by several factors, including crop type, soil properties, water body characteristics (such as depth and flow rate), land use, slope, and proximity to water bodies, as well as meteorological conditions like temperature, rainfall, moisture, and wind [80]. Groundwater contamination with pharmaceuticals occurs through leaching, a process in which water infiltrates the soil and carries pharmaceutical compounds downward. Additionally, erosion can facilitate the transport of pharmaceuticals into surface waters. Water-induced erosion, along with the deposition of soil particles containing pharmaceutical residues, can lead to the contamination of adjacent water bodies [59,81].

#### 2.2.2. Microplastics

Agricultural runoff is a significant source of microplastics in aquatic environments [82]. Plastic films are commonly used as mulch by farmers to suppress weed growth, conserve moisture, and increase soil temperature [84]. Many fertilizers and pesticides are packaged in plastic-based materials, which, when improperly disposed of or subjected to natural degradation through sunlight, mechanical wear, and microbial activity, release microplastics into the soil. Following rainfall, storms, floods, or irrigation, these plastic particles are transported into aquatic ecosystems via runoff. Soil erosion, prevalent in agricultural areas, further contributes to the transfer of microplastics to water bodies [83,85]. Additionally, wind can carry plastics from farmland to nearby aquatic systems through atmospheric transport. Moreover, plastics used in livestock farming, such as packaging for feed bags, additives, or medicinal items, can degrade over time, leaching microplastics into surrounding aquatic environments.

Agricultural runoff is increasingly recognized as a major non-point source of aquatic microplastics (MPs), significantly contributing to plastic pollution in both freshwater and marine environments [86]. The widespread use of plastic mulch in agriculture, where thin sheets of polyethylene are laid on the surface of the soil to suppress weed growth, maintain moisture, and increase the temperature of the soil, thus increasing yields [87]. The film undergoes physical and chemical degradation over time as a result of long-term exposure to sunlight (UV light), tillage, microbial activity, and temperature fluctuations, leading to the fragmentation of plastics into micro-sized particles [88]. These fragments of plastic can accumulate in soil and eventually be mobilized into the surrounding water systems by surface runoff, particularly after heavy rainfall, flood events, or irrigation practices [86,89]. In addition to the mulch film, agricultural chemicals such as fertilizers and pesticides are often packed in plastic containers or in polymerized pellets, which may release MP if they are disposed of incorrectly or if they are gradually weatherized [90]. Soil erosion, particularly on slopes and in poorly managed agricultural lands, is a primary mechanism for the transport of microplastics (MPs), which are carried into rivers, lakes, and estuaries during runoff events [86]. In addition to waterborne transport, atmospheric dispersal plays a crucial role; wind can lift lightweight plastic particles from open fields, carrying them through the air to nearby water bodies or causing them to accumulate in watercourses [90]. Furthermore, plastics used in animal husbandry, such as packaging for feed, silage, medical supplies, and equipment, can also contribute to MP pollution. Improper disposal or degradation due to prolonged use and exposure to weathering further exacerbates this issue [91].

### 2.3. Aquaculture Operations

#### 2.3.1. Pharmaceuticals

Aquaculture effluents have the potential to release pharmaceuticals used in aquaculture, such as antibiotics and antiparasitic agents, into freshwater environments [92]. Pharmaceutical pollution in aquaculture systems can also result from improper disposal of medicated feed [39]. Pharmaceuticals from aquaculture operations are primarily introduced into freshwater ecosystems through the use of veterinary pharmaceuticals, which are commonly administered as bath formulations or medicated feed [93]. Antibiotics, antiparasitic drugs, and disinfectants are extensively used in aquaculture facilities to prevent and treat diseases among fish populations [94]. Veterinary pharmaceuticals are typically administered to fish via injection, bath treatments, or medicated feed [93]. Fish treated with these pharmaceuticals often excrete residual medications, which, through aquaculture effluents, can enter nearby water bodies [95]. If not properly managed, pharmaceutical waste can contaminate freshwater ecosystems, potentially causing significant ecological impacts. Adopting the “reconciliation ecology” paradigm for freshwater ecosystem management will be essential to mitigate such risks [96]. Drugs are widely distributed and can inadvertently find their way into freshwater ecosystems through aquaculture effluents. Pharmaceuticals and their metabolites are not the only mixtures of chemicals found in aquaculture facilities’ effluents. Other pharmaceutical compounds include disinfectants, diagnostic agents, antibiotics, and antiparasitic agents [48]. Usually, aquaculture effluents are dumped into neighboring bodies of water, like lakes, rivers, or coastal waters. High levels of pharmaceuticals in aquaculture effluents may have an adverse effect on aquatic life and water quality in the receiving environment [5,60,62,92,97].

#### 2.3.2. Microplastics

Aquaculture practices are identified as a significant contributor of MPs to the aquatic systems in a number of ways [98]. Plastic materials are extensively used in both aquaculture and mariculture operations, and these activities continue to be a significant source of plastic litter, even in marine environments [99]. Aquaculture-related plastics have been detected in mariculture areas and the surrounding waters [98]. Commercial feeds, commonly used in aquaculture, may contain microplastics (MPs) as impurities in the raw materials. Coastal aquaculture operations can thus be significant contributors to plastic litter in coastal waters [99]. Different plastic-based infrastructures and equipment such as nets, buoys, and ropes are frequently used in aquaculture in addition to the packaging and showcasing of the final products in the value chain [100,101]. The intended or accidental disposal of these plastic products could serve as a source of MPs in water. Mariculture areas and nearby waters have been found to contain plastics related to aquaculture [102]. Additionally, residues of synthetic polymers have been discovered embedded in the sediment surrounding aquaculture facilities, suggesting the possibility of accumulation and long-term environmental persistence [103]. Commercial feeding stuffs, which are common items used in aquaculture, may contain micro-organisms present as impurities in the raw material. Moreover, the pelleting process used in the production of aquafeed may unintentionally contain micro-organisms via contaminated processing equipment or packaging material [104]. One of the main causes of plastic debris in coastal waters may be coastal aquaculture [103].

Aquaculture commonly uses various plastic-based infrastructures and equipment, including nets, buoys, and ropes, in addition to packaging and displaying the end products along the value chain [104]. The repetitive use of plastic-based equipment, combined with harsh environmental conditions such as waves and salinity, can accelerate the fragmentation of these materials, contributing further to microplastic release [105]. The deliberate or accidental disposal of such plastic products could act as a water table damper. Furthermore, the loss or disappearance of aquaculture gear, often referred to as ghost gear, remains a major problem, as the lost plastic is still degrading into MP and poses a threat to marine life [102,105].

### 2.4. Land Application of Biosolids

#### 2.4.1. Pharmaceuticals

Organic materials obtained from sewage sludge, known as biosolids, are frequently utilized as fertilizer in agriculture [47]. Antibiotics used in veterinary care are dispersed as an organic fertilizer onto agricultural land in the form of biosolids [106,107,108]. Pharmaceuticals found in biosolids have the potential to seep into soil and groundwater, posing a risk of contaminating surface waters via runoff [109]. Biosolids, nutrient-rich organic materials made from sewage sludge, are applied to land, which allows pharmaceuticals to enter freshwater ecosystems. To increase soil fertility and crop yields, biosolids are frequently spread as fertilizer to agricultural lands [107,108]. In addition to other contaminants from household and industrial sources, biosolids may contain pharmaceutical residues from human excretion [50]. These drug residues may remain in biosolids even after wastewater treatment facilities have finished treating them [56]. Pharmaceutical residues may eventually find their way into surface waters through runoff or infiltration into the soil and groundwater when biosolids are applied to agricultural lands. In freshwater ecosystems, then, one major source of pharmaceutical pollution may come from the use of biosolids as fertilizer. Biosolids have the potential to enter surface waters via leaching and runoff [110]. Additionally, erosion can carry pharmaceuticals into surface waters [59,81]. Water erosion and subsequent deposition of soil particles containing pharmaceutical residues into adjacent water bodies can result in pharmaceutical contamination.

#### 2.4.2. Microplastics

Plastic products are used by humans mostly in terrestrial environments, and so the wastes are piled up primarily on land, which seeps into the aquatic systems by runoff during rain, storms, and floods over there and finally gathers into the oceans [111]. According to recent estimates major share of marine plastic litter originates from human actions performed on land [112]. Between 4.8 and 12.7 million metric tons of plastic in the oceans today are believed to be sourced from terrestrial environments. The more worries lie in the fact that this share has a good chance of rising in the coming decades [100]. Plastic input into the environment is further increased by roadside litter and agricultural plastic debris, such as mulch films and fertilizers coated in plastic, which decompose into microplastic fragments [113]. Due in large part to inadequate solid waste management, illicit dumping, and industrial leakage, plastic pollution has an impact that goes well beyond terrestrial ecosystems despite coming primarily from land-based activities [114]. Through direct discharge, river transportation, and extreme weather events like hurricanes and monsoons, plastic waste enters aquatic systems [115]. Unless preventive measures are taken, estimates indicate that plastic input into oceans will increase, potentially exacerbating ecological and socioeconomic consequences, including harm to marine organisms and financial losses in fisheries and tourism [116]. The situation is further exacerbated by rising plastic production, poor waste disposal, and climate change-driven factors like flooding and extreme precipitation, which speed up the movement of plastic from land to sea [113,116].

### 2.5. Atmospheric Deposition

#### 2.5.1. Pharmaceuticals

Precipitation and atmospheric fallout have the ability to carry pharmaceuticals through the atmosphere and deposit them alongside freshwater bodies [56,117]. This procedure may lead to the contamination of distant or pure freshwater environments with pharmaceuticals [50,59,74,118]. Medicinal compounds in the atmosphere can settle on land and in water surfaces through a process known as atmospheric deposition, and this allows pharmaceuticals to find their way into freshwater ecosystems [48,75,119]. Emissions from numerous human activities, such as transportation, agriculture, and industrial processes, are the primary sources of pharmaceuticals in the atmosphere [120]. These actions discharge medicinal substances into the atmosphere, where wind and atmospheric currents are responsible for long-range transport [121,122]. Pharmaceutical residues that have been volatilized from drinking water are another source of pharmaceuticals in the atmosphere [109]. Pharmaceuticals, for instance, that are sprayed on crops or dumped into surface waters may evaporate into the atmosphere and cause atmospheric deposition [123]. The reason for their widespread use is that, once released into the atmosphere, pharmaceuticals can travel great distances and deposit themselves on land and in water through an atmospheric wet deposition process. Wet deposition occurs when pharmaceuticals are dissolved in rain or snow and then deposited onto surfaces [124].

#### 2.5.2. Microplastics

Once released into the environment from various sources, microplastics (MPs) can be transported through the air due to their lightweight nature, enabling them to travel long distances via atmospheric transport. These atmospheric MPs can then fall directly into lakes, rivers, and oceans through both dry and wet deposition [125]. The microplastics deposited on land can eventually enter aquatic systems through runoff. Field-based research has revealed that atmospheric microplastics were a significant source of marine microplastic pollution [126]. Snowfall has been reported to capture a greater diversity of microplastic sizes and shapes compared to rainfall. Studies have presented both quantitative and qualitative compositions of microplastics deposited from the atmosphere in coastal zones, highlighting the link between microplastic deposition and meteorological factors [127,128].

After being released into the environment from various sources, microplastics (MPs) can be transported through the atmosphere due to their lightweight nature, allowing them to travel over long distances. Studies have shown that MPs can remain suspended in the atmosphere and undergo long-range transport, eventually reaching remote areas far from their original sources [129]. These airborne pollutants can be directly deposited into lakes, rivers, and oceans through both dry and wet deposition. Dry deposition occurs when airborne microplastics settle to the ground without precipitation, while wet deposition refers to the removal of microplastics from the atmosphere through precipitation events, such as rain or snow [130]. Unsuitably disposed waste is carried by runoff into drainage systems and water, which is one of the main causes of plastic pollution in urban areas with inadequate waste man-agement systems [131,132]. Microplastics (MPs) deposited on land can eventually enter aquatic systems through runoff, often facilitated by vegetation. For instance, microplastics deposited on urban surfaces may be washed into watercourses during rainfall events, thereby contributing to water pollution [133]. Field studies have shown that microplastics in the atmosphere are a significant source of marine microplastic pollution, highlighting the crucial role of atmospheric pathways in the global distribution of microplastics [127]. Snowfall has been found to account for greater variability in microplastic (MP) sizes and shapes compared to rainfall. This is likely due to the larger surface area and more complex structure of snowflakes, which enable them to capture a broader range of MP particles from the atmosphere [128,133]. The studies also provided quantitative and qualitative data on the composition of microplastics deposited in coastal zones from the atmosphere and the relationship between the deposition of MP and meteorological factors. It was found that factors such as wind speeds, humidity, and precipitation patterns affect deposition rates and atmospheric MP characteristics in these regions [128].

### 2.6. The Origins and Persistence of Microplastics and Pharmaceuticals in Aquatic Environments

Aquatic environments frequently contain pharmaceuticals and microplastics because of their frequent use, inappropriate disposal, and resistance to degradation. Their presence is primarily attributed to the following:

Wastewater Treatment Plants (WWTPs): Municipal and industrial wastewater are the source of many pharmaceuticals and microplastics. These contaminants can enter surface waters because conventional WWTPs are not built to completely remove them [131,132];

Industrial and Agricultural Runoff: Surface runoff and drainage systems can allow pharmaceuticals used in veterinary care, as well as microplastics from industrial processes, to enter water bodies [134];

Household Contributions: Wastewater systems are exposed to pharmaceuticals and microplastic fibers when synthetic textiles are washed, and unused pharmaceuticals are improperly disposed of [135];

Persistence and Environmental Conditions: These contaminants’ chemical stability, resistance to biodegradation, and capacity to adsorb onto sediments, which lengthens their lifespan and increases their potential for bioaccumulation, allow them to persist in aquatic environments [135].

### 2.7. Environmental and Toxicological Effects: Disadvantages

Pharmaceuticals and microplastics in aquatic ecosystems pose severe environmental and health risks:

Bioaccumulation and Biomagnification: These pollutants can pose long-term ecological and health hazards because they build up in aquatic organisms and become more severe as they go up the food chain [135];

Toxicity to Aquatic Life: Fish and other aquatic species have been shown to exhibit altered behavior, developmental abnormalities, and decreased fertility as a result of pharmaceuticals and microplastics [132];

Promotion of Antimicrobial Resistance (AMR): A serious risk to the environment and public health, antibiotic-resistant bacteria are a result of persistent pharmaceutical residues in water bodies [131];

Endocrine Disruption: Reproductive disorders like the feminization of fish populations are caused by pharmaceuticals that disrupt endocrine systems, especially hormones [135];

Limited Removal Efficiency: Due to the limitations of conventional wastewater treatment plants (WWTPs), these pollutants are continuously released into natural water bodies [136];

Potential Human Health Risks: Long-term exposure to microplastics and pharmaceutical residues is a concern when contaminated seafood and drinking water are consumed, but more research is required to completely comprehend these effects [135].

Recent research on the bioaccumulation of pharmaceuticals and microplastics in various aquatic organisms is summarized in Table 3, which highlights the ecological ramifications of these new pollutants. In fish species like *Perca fluviatilis*, pharmaceutical compounds like diphenhydramine, oxazepam, and hydroxyzine have been found, but trimethoprim and diclofenac were not. Furthermore, a broad spectrum of microplastic particles has been measured in a variety of freshwater and marine species, with concentrations ranging from 0.33 to 6.71 particles per individual. The table attempts to highlight how these pollutants are absorbed and accumulate biologically, as documented in earlier studies.

## 3. Effects on Freshwater Fish

The issue of pharmaceuticals in the environment has been extensively addressed by numerous authors, with approximately 18,000 documents available on the subject, the majority of which are published scientific studies [141]. Worldwide, about 3000 structurally different pharmaceuticals are regularly used. The majority of the rivers in the world contain many. Exposure to active pharmaceutical ingredients (APIs) in the environment can have detrimental impacts on human and ecological health [142]. Two facts were established over 20 years ago: first, that human pharmaceuticals were undeniably present in aquatic environments, and second, that there was a significant likelihood that some of these substances could exist in concentrations harmful to certain aquatic organisms [143].

### 3.1. The Bioaccumulation and Biomagnification of Pharmaceuticals Within Freshwater Food Chains

According to Meador and Miller et al.’s definition, bioaccumulation refers to the simple uptake of substances from the environment or their gradual accumulation or retention [138,144]. In other words, when an organism’s absorption of a pollutant surpasses its capacity for digestion, bioaccumulation takes place [145]. Over 200 neuroactive pharmaceuticals are currently being used in clinical settings, and a significant portion of these drugs (*n* = 84) have been found to be found in rivers all over the globe [146]. Numerous of these latter substances are expected to bioaccumulate in fish because they are comparatively hydrophobics. Pharmaceuticals can also accumulate in fish tissues and increase in concentration as they move up the food chain, leading to higher levels in predatory fish species [147]. Fluoroquinolones (FQs) have been found to accumulate more in organisms with higher lipid contents; significant concentrations of FQs have been found in aquatic organisms’ tissues, including fish, as well as in surface waters across the globe [148]. The biota-sediment accumulation factor (BSAF) is a useful parameter for understanding the partitioning of pharmaceutical contamination from sediment to benthic organisms. Thus, investigation into the BSAFs of pharmaceuticals in benthic organisms will improve our understanding of how pharmaceuticals enter the aquatic food web. The typical method for calculating bioaccumulation factors is to compare the concentration of the compound of interest in the biota sample (plants, animals) to that in the surrounding media (either in the soil or in the water), including BSAF [149]. However, pharmaceuticals do not currently have access to field-based BSAF data [150]. It has been observed that pharmaceuticals are pseudo-persistent as a result of their constant discharge into water bodies [150,151]. In ecological risk assessments, bioaccumulation and biomagnification are two key ideas that are used to quantify the amount of pollutant transport within food webs [152]. Hence, the term “biomagnification” in relation to a food web refers to the rise in a contaminant’s concentration in one organism relative to that of its prey, such as microplastics or pharmaceuticals [153]. A specific biomagnification pattern has been noted, however, for several antibiotics: norfloxacin and enrofloxacin [154], diclofenac [155], roxithromycin [156], and ciprofloxacin [150].

According to a recent study, the COVID-19 pandemic has made the issue of pharmaceutical and personal care product (PPCPs) accumulation in the environment more pressing because of the increased use of disinfectants and other products [157]. Pharmaceuticals have been shown to have pseudo-persistent qualities in surface waters that receive effluents discharged from wastewater treatment plants, which has led to their bioaccumulation by non-target organisms like fish [158,159,160]. At the same time, more and more species from both inland and coastal aquatic systems are being found to contain pharmaceuticals [161]. Drugs do not usually biomagnify, as evidenced by the possibility of trophic transfer from freshwater systems at lower latitudes [137,150,156,162]. The Eurasian perch (*Perca fluviatilis*) and the dragonfly larvae (*Aeshna grandis*), two freshwater predatory species, were found to have higher concentrations of the anxiolytic oxazepam in comparison to their food, according to a study that did not find evidence of trophic transfer of other compounds [163]. When pharmaceuticals bioaccumulate in non-target organisms, such as surface waters, they have the potential to enter the food chain.eg: biota (aquatic and riparian) [164]. According to a recent study, the Arctic food web demonstrates how stimulants and pharmaceuticals behave differently depending on the target compound. Thus, inter-compound variation may occur during the trophic transfer of these compounds [165]. The development of antibiotic resistance, interference with biochemical processes, endocrine system disruption, bioaccumulation of pharmaceuticals in non-target organisms, and other direct and indirect effects are just a few of the risks that pharmaceutical compounds can have [166,167]. Nowadays, pharmaceuticals can now be found in all areas of the environment.

### 3.2. Bioaccumulation and Biomagnification of Microplastics in Aquatic Food Chain

Microplastics may accumulate in organisms and multiply along the food chain, causing higher quantities in predatory species [168,169]. The concentration of these contaminants can rise when larger fish and marine mammals eat smaller creatures tainted with microplastics [170]. For example, research has shown that large concentrations of microplastics can build up in the tissues of predatory species, such as swordfish and tuna. Microplastics could also enter the food chain by possibly being integrated into marine aggregates [171]. This transfer can result in increased amounts of microplastics in larger species, a process known as biomagnification. Microplastic biomagnification has the potential to destabilize aquatic ecosystems, impacting population dynamics, species composition, and reproductive success [168]. The stability of the entire ecosystem may be disrupted when important species are impacted.

## 4. Physiological Effects on Fish, Encompassing Effects on Growth, Reproduction, Immune System Performance, and Behavioral Modifications

### 4.1. Pharmaceuticals

#### 4.1.1. Immune System and Toxicity Effects

Numerous tons of chemical and pharmaceutical materials are produced and used annually throughout the world. One significant category of newly discovered environmental micropollutants is pharmaceuticals. However, the majority of these drugs have the potential to degrade either biotically or abiotically, accumulating in the tissues of fish and other aquatic organisms to cause unwanted behavior, histopathology, interference with reproduction, and immunotoxic reactions, among other possible toxicological effects [172]. Changes in behavior and variation are crucial for individual performance [173,174], species evolution [175], and ecosystem function [176]. Changes in behavioral patterns, histological modifications, biochemical parameter changes, or other physiological markers can all be used to see these effects. The well-discussed effects of these environmental pollutants on human health include their potential to disrupt hormones, influence brain development and function, and have a common effect on human health [177,178]. Antidepressants are also a significant concern, as serotonin levels impact both physiology [179]. Fish can experience acute or chronic damage from exposure to pharmaceuticals. Acute damage occurs suddenly and may lead to tissue damage, organ failure, or death, while chronic damage develops over time and can have long-term effects on the health and survival of fish [180,181]. The behavioral changes, personal health, nourishment, development, and survival, and subsequent changes to demographic variables like birth, death, and migration rates [182] may have a cascading effect on ecosystem functioning and population dynamics. This type of contaminant can have toxic effects on non-target organisms. The research assessed tetracycline’s acute toxicity in several species of freshwater fish [183]. Identification of histological alterations in the liver and gills, modifications in antioxidant protection levels (including GST, CAT, and lipoperoxidative damage), and assessment of potential neurotoxic effects (such as acetylcholinesterase activity) [183].

Tetracyclines, which are broad-spectrum antibiotics, are frequently and extensively used in veterinary and human medicine. Previous studies have demonstrated the effects of tetracycline on the catalase (CAT) activity of living organisms, in addition to having the capacity to cause oxidative damage [184] and phytotoxicity in creatures like mammals and earthworms [185,186]. The results indicate a potential causal link between exposure to tetracycline and changes in histology, specifically in gills, as well as enzyme activity in the liver and gills. This suggests that tetracycline may have pro-oxidative effects [183].

Another significant change in fish health is the alteration of immune function caused directly by toxic compounds [172]. Milla et al.’s research study of 63 participants revealed the impact of synthetic steroids on fish immune systems, including both androgenic and estrogenic steroids [187]. In a study conducted by Liang et al., it was found that exposure to norfloxacin nicotinate (NOR-N), an antibacterial fluoroquinolone, led to an increase in abnormality and mortality in the early life stages of zebrafish (Danio rerio), as well as a decrease in hatching rate and body length [188].

A marked decrease in immune system function was noted in harbor seals (Phoca vitulina) reduced lymphocyte transformation and the G0/G1 phase of the cell cycle, as indicated by exposure to a combination of 17α-ethinyl estradiol and 25,000 μg/L naproxen [189], endocrine disruptive and immunomodulation activities [190]. Research has demonstrated that FQs have an impact on aquatic plant growth and development as well as the antioxidant defense system [148]. Researchers have discovered that when young zebrafish (Danio rerio) were exposed to benzotriazole ultraviolet stabilizers for 28 days, they developed immunotoxic reactions that were correlated with liver damage (inflammation, hepatic vacuolization, and nuclei pyknosis) [191]. The experimental findings of Bera showed in catfish exposed to triclosan, Pangasianodon hypophthalmus, triclosan reduced respiratory burst activity (RBA), myeloperoxidase activity (MPO), and phagocytic activity (PA). This suppression of both cell-mediated and humoral immune responses was observed [192]. There is mounting proof that these substances affect fish immunity, and these pharmaceutical residues pose a threat to public health and the ecological balance. Because the compounds may have additive and synergistic effects, the ecotoxicity of a single compound is lower than that of a mixture [193]. These factors influence the bioaccumulation and metabolism patterns of fluorescent queries (FQs) in aquatic organisms, as well as their ecological toxicity [148].

#### 4.1.2. Growth and Behavior

Any modifications or deviations from the typical operation of an organism’s bodily systems or processes are referred to as physiological effects. Any alterations to fish’s regular physiological processes, such as growth, reproduction, metabolism, immune system performance, and general wellbeing caused by exposure to pharmaceutical substances, would be considered physiological effects. Fish behavioral responses have been documented in the past, and examples include how they affect socialization, aggression, reproduction, predator avoidance, and learning and memory [194]. Behavioral responses for aquatic toxicity testing have drawn attention recently. They found studies on development, reproduction, acute lethality, and behavior [195]. Changes in anxiety levels may lead to modifications in ecologically significant behaviors like boldness, exploration, and activity. The physiology and behavior of non-target species may be affected by fluoxetine [170]. The behavioral changes, personal health, nourishment, development, and survival, and subsequent changes to demographic variables like birth, death, and migration rates may have a cascading effect on ecosystem functioning and population dynamics [196]. These behaviors are crucial for an individual’s fitness and play a role in various essential processes such as dispersal [182,196]. There are behavioral endpoints for several psychiatric drugs in human medicine, which may indicate similar effects in exposed wildlife [197]. A study conducted on perch from a natural population revealed that a low concentration of oxazepam to 1 μg/L, a benzodiazepine, caused exposure to cause both decreased sociality and increased activity, while high μg/L caused increased boldness [198].

Fish exposed to pharmaceuticals can experience significant disruptions in their growth patterns. Prolonged exposure may lead to impaired growth, as pharmaceuticals can disrupt normal cell proliferation and inhibit the division of cells in fish tissues. This disruption affects tissue maintenance, repair processes, and developmental pathways, which may result in structural abnormalities or impaired physiological functions [199]. There is significant evidence showing that chemical contaminants can affect the behavior of both wildlife and humans.

According to the extensive research conducted by Grzesiuk et al., it has been found that even small amounts (ng/L) of medication can have a significant impact on aquatic organisms in the long term. This study clearly found that persistent exposure to pharmaceuticals (propranolol, ibuprofen, and fluoxetine) for 30 generations on Acutodesmus obliquus and Nannochloropsis limnetica resulted in decreased cell number, increased carotenoid-to-chlorophyll ratio, and altered consumer feeding [200]. Changes in behavior have complementary effects on neurotoxicity, making them the early indicators of toxicity [201]. Studies dating back to the early 1900s have documented changes in swimming patterns in fish when exposed to different chemicals [202,203] with various studies reporting comparable effects having emerged over the past per [195,204].

Incorporating behavioral effects into chemical ecotoxicity testing has garnered significant attention recently [182,197]. Psychoactive drugs, in particular, antidepressants, have been shown by numerous scientists to impact different aspects of behavior in a variety of aquatic organisms. This is despite the unquestionably difficult task of first collecting and then interpreting behavioral data [126]. Fish behavior can be affected by pharmaceuticals, which could change their typical patterns of activity, feeding, mating, or social interactions [152]. The functioning of ecosystems and population dynamics may be impacted in a cascade manner by these behavioral changes [187,188], personal health, nourishment, development, and survival [182] and consequently cause changes to demographic variables like the rates of birth, death, and migration [182].

#### 4.1.3. Reproduction

Chemicals have been classified as posing dangers to freshwater biodiversity, putting it under greater threat [205]. Numerous studies have shown that common aquaculture practices, such as capturing wild fish to harvest gametes, fostering social interactions at artificial stocking densities, and performing routine husbandry tasks like handling and confinement, are stressful to fish and may have an adverse effect on their ability to reproduce and grow, which in turn compromises their immune system [206], that is why major worldwide attention has focused on the potential for endocrine-disrupting chemicals (EDCs) to cause reproductive system disruption [207,208,209,210]. Fish reproduction is affected by prolonged contact with pharmaceutical concentrations in the environment; research on the effect on reproductive success and the mechanism of disruption revealed little evidence as predicted. Reduced fecundity and competitive population failure without fertilization following an extended period of exposure were among the effects [211]. This altered their morphology and physiology, leading to numerous gland-related issues, intersexuality being particularly prevalent [212,213], which is the induction of proteins unique to females in fish males [209], and imbalanced masculinity relationships, which probably have adverse effects on a community [211,212]. Decreased sperm counts affect its traits and behavior [214,215]. The estrogenic potency of some EACs has raised concerns about the adverse impact they could have on the breeding and survival of wild animal populations [212]. Steroidal estrogens, such as estrone (E1), estradiol (E2), and synthetic estrogen EE2, play a significant role in controlling sexual differentiation and development. These hormones are potent regulators of both sexual development and reproductive capacity [216,217,218]. At least one human pharmaceutical, ethinylestradiol (EE2), was also shown to have dramatic negative effects on fish reproduction when it was present in the water at very low concentrations more than 20 years ago [219]. Antidepressants are also a significant concern behavior in a variety of creatures, fish included [220,221], and contribute significantly to activity levels, aggression, and reproductive behaviors [221,222]. Fluoxetine has the potential to impact the behavior and physiology of non-target species [214], interrupt the process of reproduction [223], characteristics of sperm in fish [224], and movement between the lake and the adjoining streams [225]. The significant impact of prolonged exposure to mixed pharmaceutical substances on stream organisms is often overlooked. Fish behavior may be affected by pharmaceuticals, leading to changes in their activity, feeding habits, reproductive patterns, and social interactions. These alterations in behavior could have far-reaching consequences on the state of each individual organism, population dynamics, and ecosystem function as a whole [226,227]. Fish may experience disruptions in their reproductive systems when exposed to pharmaceuticals, leading to potential issues in successfully reproducing. These disruptions may present as decreased fertility, compromised egg or sperm quality, changes in spawning behavior, or abnormalities in the development of offspring [172,224,228]. An investigation says the effects of exposure to common drug residues have been noted (carbamazepine (CBZ)) on four generations of zebrafish, including decreased reproductive function, courtship behavior, aggression, sperm speed, and morphology [229]. Palace et al. conducted two studies, one in 2006 and the other in 2009. Both studies revealed that all males who were exposed to certain factors exhibited delays in spermatocyte development. Additionally, intersex conditions were found in approximately one-third of the males [230,231]. Adult fathead minnows’ sperm parameters decreased when exposed to the human medication clofibric acid, according to Runnalls’ research [232]. Fish exposed to pharmaceutical effluent downstream of the Dore River in France showed altered enzyme activity, neurotoxicity, intersex traits, and vitellogenin production, according to in situ studies [233]. Another research has demonstrated that specific pharmaceutical contaminants found in the environment can affect the reproduction of fish through the serotonin system [234].

Another study discovered a decline in reproductive performance and success rates in zebrafish exposed to ibuprofen at concentrations commonly found in the environment [235]. A study discovered a notable decrease in reproductive functions in male Astyanax altiparanae fish when exposed to standard levels of common drugs diclofenac (DCF) and caffeine (CAF), which resulted in lower levels of 17β-estradiol (E2) and testosterone [236]. Liang et al.’s recent laboratory study revealed that males exposed to environmentally relevant concentrations of 3-(4-Methylbenzylidene) camphor (4-MBC) saw decreased spermatogenesis, decreased plasma 11-ketotestosterone levels, increased reproductive toxicity, and anti-androgenicity in Japanese medaka (*Oryzias latipes*) [237]. De Lima and colleagues discovered that specific diets aimed at lowering oxidative stress in humans might disrupt reproductive processes and development in female Oreochromis niloticus tilapia (Niloticus) [238]. Studies have shown that pharmaceuticals present in aquatic environments can have adverse effects on the reproductive functions of fish. This is evident through changes in sperm parameters, vitellogenin induction, intersex traits, and enzyme activities in fish exposed to these substances.

#### 4.1.4. Hospital Wastewater and DNA Damage

Several pharmaceuticals used in hospitals (e.g., antibiotics and cytostatic drugs) give rise to further worries regarding the possible risk that hospital wastewater discharge poses to people and the environment by damaging the DNA of bacteria or eukaryotic cells [239].

### 4.2. Microplastics

#### 4.2.1. Physiological Disorders in Fish Due to Microplastics (MPs)

Fish suffer from various physiological disorders brought on by MPs, such as oxidative stress, neurotoxicity, and immunotoxicity [240,241]. Fish species may experience reproductive difficulties due to physiological disturbances, which could affect population sizes [221]. These disturbances can compromise the overall health and survival of fish, leading to reproductive difficulties, which could have significant impacts on population sizes. If these physiological disorders persist, they may result in long-term declines in fish populations, altering ecosystem dynamics and biodiversity.

#### 4.2.2. Health Impacts of MPs on Fish

Fish ingesting microplastics may suffer from serious health problems such as inflammation, decreased feeding intensity, digestive tract blockages, impaired gill performance, immunosuppression, and hampered reproduction [240,242]. These health issues can further affect the fish’s reproductive capabilities, leading to lower birth rates and fewer healthy offspring.

#### 4.2.3. Specific Impact on Freshwater Fish

Researchers demonstrated that polylactic acid MPs affected the growth performance, induced considerable changes in body proximate composition, alteration in the blood profile, increased intestinal abnormalities, and fall of mineral content in the muscles of freshwater fish, *Cirrhinus mrigala* [243]. When exposed to MPs, the fish experienced changes in body composition, particularly in the distribution of nutrients, which may affect their overall fitness and survival. Alterations in the blood profile were also noted, which could indicate physiological stress and changes in metabolic processes. Additionally, the fish displayed increased abnormalities in their intestines, possibly due to blockages or inflammation caused by the MPs. The study also found a decrease in the mineral content in the fish’s muscles, which could weaken the fish and make it more vulnerable to predators or further environmental stress.

#### 4.2.4. Accumulation of MPs and Organ Dysfunction

The harmful effects on fish health are exacerbated by the accumulation of MPs, which also interferes with the liver and kidneys’ regular functions [244]. The liver, in particular, plays a crucial role in metabolism and detoxification, so its impairment can have cascading effects on the fish’s overall health. Similarly, the kidneys are responsible for regulating waste and fluid balance, and their dysfunction could lead to dehydration or toxicity buildup in the fish’s system, further reducing its chances of survival.

#### 4.2.5. Broader Impact of MPs on Fish Health

MPs can cause oxidative stress, impaired metabolism, immunological responses, and organ function, which can lead to cellular damage, including the deterioration of lipids, proteins, and DNA [245]. These cellular damages can accumulate over time, weakening the fish and increasing its vulnerability to diseases and environmental changes.

## 5. Impacts on Fish

### 5.1. Ecological Effects of Pharmaceuticals on Fish Populations

Fish feeding, mating, and predator avoidance behaviors can change after prolonged exposure to microplastics and pharmaceutical pollution, which may have an effect on population dynamics and community structure [246,247]. The sustainability of resident fish populations may be impacted by fish population declines, and trophic cascades may indirectly alter other taxa [231]. Recently, pharmaceuticals have been found in nine of the fourteen sites where drinking water has been sampled. Under South Florida’s subtropical climate, pharmaceutical use and its ecological effects are likewise restricted [248]. Pharmacological effects on behavior are ecologically significant because behavior is closely linked to both individual fitness and population persistence [175,249]. It is true that some behaviors have a direct impact on fitness; however, in addition to these direct effects, modifications in personal fitness may also have indirect ecological effects. Changes in species interactions, like predation or competition, result in these indirect effects [250]. For instance, when personal habits shift, several compromises alter personal fitness and can cause a change in population size or even local extinction [251]. The remaining community suffers when a species goes extinct, but population size fluctuations can also affect population dynamics or food-web cascades that follow, say, a rise or fall in the feeding efficiency of a pharmaceutically exposed species [197]. Fish have become more feminine as a result of oral contraceptives, and the issue of antimicrobial resistance is made worse by the overuse and accidental release of antibiotics into waterways [252]. Additional indirect ecological effects include alterations in species richness and community composition that follow population size changes (particularly extinctions), as these are known to affect ecosystem functioning [246,253]. These could be particularly likely if various taxa react differently to pharmacological exposure [197]. Numerous pharmaceutical groups have been found to affect a variety of behaviors that are crucial for ecosystem functioning, food-web properties, and fitness [197]. Stressors caused by pollution may also cause changes in fish abundance and distribution, which could have an impact on ecosystem functioning and species composition [254,255]. Antidepressants have been shown to cause starlings to eat less, and contraceptive drugs have been shown to reduce fish populations in lakes. These findings suggest that drugs that are flushed into the environment may be the cause of wildlife decline. Pharmaceuticals have the potential to have significant effects on ecosystems and wildlife because they are used in thousands of cases worldwide [256].

Another study found that starlings fed less frequently during the prime foraging periods of sunrise and sunset when exposed to the common antidepressant fluoxetine at the low levels expected in the environment. Crucially, it should be noted that fluoxetine is not the sole antidepressant found in the environment or even the only pharmaceutical [257]. Another study revealed that the synthetic estrogen found in birth control pills severely disturbed the ecosystem as a whole, in addition to eradicating fathead minnows from lakes used for experimentation in Ontario. The loss of the minnow and other prey caused the top predator in the lakes, the trout, to drop by 23–42%, while the number of insects increased because the minnows were no longer consuming the insects [246]. In addition to having an indirect negative impact on public health, these residues typically harm both targeted and non-targeted aquatic organisms [258]. Human consumption of these fish may expose people to these residues, which may have negative health effects [252]. The pharmaceuticals used for human health, hormones, antibiotics, analgesics, antidepressants, and anticancer drugs, as well as the veterinary pharmaceuticals, hormones, antibiotics, and parasiticides, have been shown to have unfavorable effects on ecosystems, including mortality. These latter categories are of particular concern [252].

### 5.2. Ecological Effects of Microplastics on Fish Populations

There are worries over the availability and potential hazards to aquatic biota due to the high frequency of microplastics in aquatic habitats. Fish often mistake microplastics for food, altering their foraging behavior and leading to physical blockages, reduced feeding, and nutritional deficiencies [259,260]. The most robust evidence was discovered for increased variability in mucus secretion (intestinal impacts), hatching success (reproduction), food intake, growth, and survival rates [260]. There were repeated reports of markedly lower levels of acetylcholinesterase (AchE) activity and somewhat lower catalase function [260]. Moreover, hazardous substances from the surrounding water, such as heavy metals and persistent organic pollutants, may be drawn to and absorbed by microplastics [261]. Chemicals used in the production of plastic, including additives, and heavy metals, persistent organic pollutants, can seep from microplastics and affect marine life [261]. These poisons can penetrate the fish’s scales and body and hence endanger both their health and that of any predators, including humans, when consumed [262]. These alterations can influence survival rates and reproductive success since some research indicates that microplastics might interfere with fish reproduction, resulting in lower fertility rates and developmental abnormalities in progeny [241,260]. In addition to its potential toxin effects on wildlife, microplastics can serve as carriers of pathogens and hazardous substances [263].

Fish that consume microplastics have lower feeding efficiency and lower energy stores, and physiological stress impairs immunity, leaving them more susceptible to illness. Furthermore, microplastics impede population growth, disrupt reproductive processes, and build up throughout the food chain, all of which exacerbate their effects on higher trophic levels. These problems are made worse by habitat degradation, which eventually results in declining fish populations, unbalanced predator–prey dynamics, and general ecosystem instability.

Microplastics are frequently consumed by aquatic life because they are mistaken for food, which reduces feeding efficiency. According to studies, this has an impact on fish larvae and marine copepods, decreasing their feeding rates, depleting their energy stores, stunting their growth, and raising mortality rates all of which endanger fish populations. Fish exposed to microplastics experience physiological stress, which results in neurotoxicity, altered behavior, and compromised immune function, making them more susceptible to illness and increasing mortality rates.

By decreasing fertility and causing developmental problems, microplastics interfere with aquatic reproduction and cause population declines. Additionally, they damage fish populations by bioaccumulating and biomagnifying through the food chain, exposing top predators to higher concentrations. Through suffocating coral reefs and altering sediments, microplastics change fish habitats, lowering resources and fish-friendly habitats, causing population declines and ecological imbalances [264].

## 6. Impact on Human Health

Pharmaceutical and microplastic contamination represents a significant and multifaceted threat to human health, primarily through exposure routes such as water consumption, dietary intake (especially seafood and fish), and inhalation. Research has consistently identified pharmaceutical residues, including antibiotics, hormones, and painkillers, in drinking water supplies across multiple continents. This widespread contamination exposes populations to chronic low-dose exposure, which may disrupt endocrine systems, weaken immune responses, and contribute to the growing global crisis of antibiotic resistance. For instance, a study by the World Health Organization (WHO) highlighted the presence of antibiotics in water systems, which can accelerate the development of resistant bacterial strains, posing a serious public health challenge [265].

Similarly, microplastics, which are pervasive in marine environments, have been found in seafood consumed by humans. These tiny plastic particles act as carriers for harmful chemicals such as bisphenols, phthalates, and heavy metals, which are known to have endocrine-disrupting and carcinogenic effects. A study published in *Environmental Science & Technology* revealed that microplastics can adsorb and transport toxic substances, increasing their bioavailability and potential harm to human health [266]. Alarmingly, recent research has detected microplastics in human blood, placental tissue, and lung samples, raising urgent concerns about their role in inflammatory diseases, metabolic disorders, and neurotoxicity. For example, a 2022 study in *Environment International* documented the presence of microplastics in human blood, suggesting their ability to travel throughout the body and potentially accumulate in vital organs [267].

The combined or synergistic effects of microplastics and pharmaceutical pollutants remain poorly understood, highlighting a critical gap in current research. Preliminary studies suggest that these contaminants may interact in ways that amplify their toxicity, but further investigation is needed to fully understand their combined impact on human health. Without immediate and coordinated intervention, these pollutants will continue to pose a significant risk to public health, particularly in vulnerable populations with limited access to clean water and adequate healthcare. Addressing this issue requires global efforts to reduce pollution at its source, improve water treatment technologies, and implement stricter regulations on plastic and pharmaceutical waste.

A systematic review of drinking water contamination by pharmaceutical residues has assessed the situation worldwide. This included 124 studies, 91 of which reported the presence of one or more pharmaceutical substances at concentrations ranging from a few nanograms per liter (ng per liter) to several tens of nanograms per liter (ng per liter). That fact underlines the global concern about the contamination of drinking water and underscores the need for close monitoring by health authorities and the scientific community [268].

The estimation of the level of microplastic contamination of seafood and the subsequent rate of human consumption was made in a study published in *Environmental Health Perspectives.* A study indicated that humans could ingest about 11,000 microplastic particles annually from the consumption of shellfish, highlighting the ingestion of shellfish as an important route of exposure [269].

## 7. Conclusions

The biodiversity and health of aquatic ecosystems are under serious threat from the increasing presence of microplastics and pharmaceuticals. Our review highlights the significant influence of these pollutants on fish physiology, behavior, and reproductive health, which can disrupt aquatic food webs and destabilize ecosystems. Despite advancements in monitoring and treatment technologies, many of these pollutants persist, contributing to chronic pollution through atmospheric deposition, agricultural runoff, and wastewater treatment processes. Given their proven capacity for bioaccumulation and biomagnification, “particularly in the case of pharmaceuticals”, further research is critically needed to understand their long-term effects on ecosystems and trophic levels.

Implementing stronger regulations, enhancing wastewater treatment technologies, and developing innovative monitoring tools are essential to mitigating the ecological risks associated with these contaminants.

The growing contamination of water systems with pharmaceuticals and microplastics is a serious and often overlooked threat to public health. These pollutants do not just disappear; they build up in the environment and move through food chains, potentially harming our immune systems and reproductive health and increasing the risk of long-term illnesses. To tackle this complex issue, we need a united, global effort that brings together experts from environmental science, toxicology, and public health to find effective solutions.

## 8. Recommendations

Reducing the effects of microplastics and pharmaceuticals in aquatic environments calls for a multifaceted strategy that includes research, policy changes, public involvement, and technology improvements.

### 8.1. Improving Wastewater Treatment Technologies

Advanced Filtration Systems: Conventional wastewater treatment plants (WWTPs) are not fully effective in removing pharmaceutical residues and microplastics. Investing in membrane bioreactors (MBRs), advanced oxidation processes (AOPs) (e.g., ozonation, photocatalysis), and nanofiltration techniques have shown promising results in removing micropollutants [51].Biodegradation Strategies: Some micro-organisms, such as Pseudomonas and Bacillus species, have demonstrated potential in breaking down pharmaceutical compounds, while enzymatic treatments could enhance MP degradation [270]. Scaling up these biological approaches could improve treatment efficiency.Regulatory Measures on Industrial and Hospital Effluents: Stricter discharge limits for pharmaceutical manufacturing plants and healthcare facilities can reduce the entry of active pharmaceutical ingredients into waterways. In Switzerland, for example, a nationwide program mandates WWTP upgrades to reduce micropollutants [271].

### 8.2. Strengthening Public Health Policies

Risk-Based Monitoring of Water Supplies: Many regions lack systematic surveillance of pharmaceuticals and microplastics in drinking water. Countries like Germany and Sweden have implemented priority substance monitoring programs, which should be expanded globally to ensure public health safety.Universal Standards for Water Safety: There is no universal guideline for pharmaceutical contamination in water. The World Health Organization (WHO) and national agencies should collaborate on setting maximum allowable concentrations for pharmaceuticals, similar to existing standards for heavy metals.

### 8.3. Raising Public Awareness and Education

Pharmaceutical Take-Back Programs: Many pharmaceuticals enter water systems due to improper disposal. Expanding drug take-back initiatives, like those in the United States (DEA National Prescription Drug Take-Back Day), can significantly reduce household pharmaceutical waste [272].Reducing Microplastic Pollution: Public awareness campaigns should highlight the impact of synthetic textiles and cosmetic products containing microplastics. Incentivizing the adoption of microplastic-free personal care products and promoting sustainable fashion choices can reduce MP emissions. The European Union’s Microplastic Restriction Initiative is an example of proactive policy action.

### 8.4. Investing in Research and Innovation

Long-Term Toxicological Studies: More research is needed to understand the chronic health impacts of exposure to pharmaceuticals and microplastics. Ongoing studies on nanoplastics and endocrine-disrupting compounds should be expanded to assess long-term risks to aquatic life and human health [273].Eco-Friendly Material Development: Alternatives to microplastics, such as biodegradable biopolymers (e.g., polylactic acid (PLA), polyhydroxyalkanoates (PHAs)), should be promoted for use in consumer products and medical applications. Investments in green chemistry approaches can lead to the development of less persistent pharmaceuticals with minimal environmental impact [274].

## Figures and Tables

**Figure 1 ijerph-22-00799-f001:**
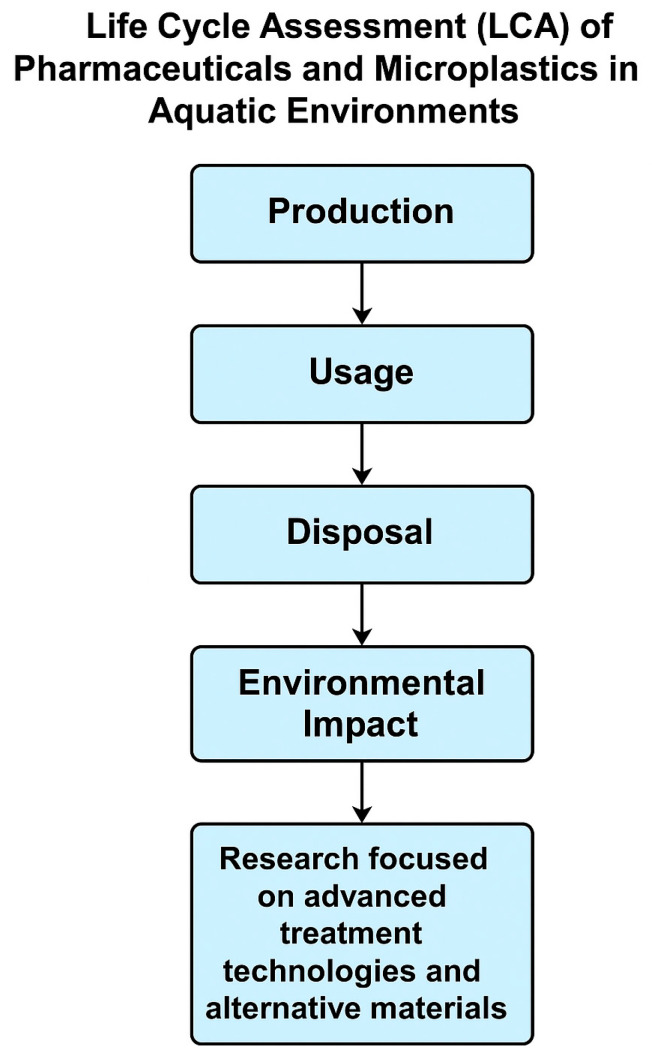
Routes from production to ecosystems for pharmaceutical contamination.

**Figure 2 ijerph-22-00799-f002:**
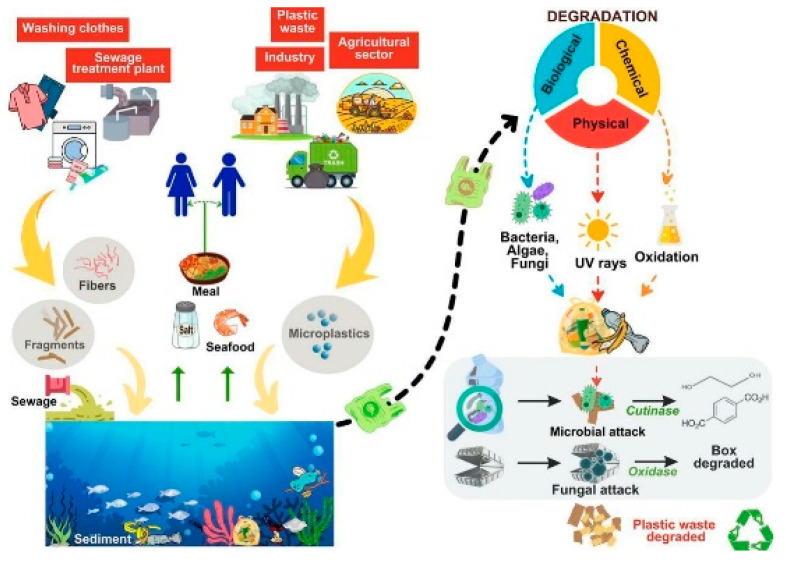
Mechanisms of environmental degradation and pathways of microplastic pollution [16].

**Figure 3 ijerph-22-00799-f003:**
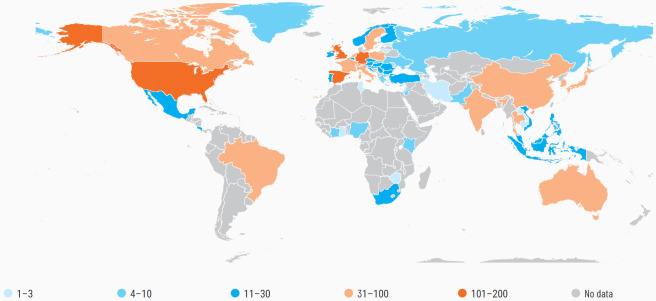
Drug distribution worldwide detected in drinking water, tap water, groundwater, and surface water [30].

**Table 1 ijerph-22-00799-t001:** Common pharmaceuticals detected in aquatic environments.

Pharmaceutical Class.	Examples	Concentration Range (ng/L–μg/L)	Source	Reference
Antibiotics	Ciprofloxacin	24.53–1491.8 ng/L	WWTP influents and effluents	[31]
	Amoxicillin	Not specified, commonly detected	WWTP effluents, hospital wastewater	
Painkillers	Ibuprofen	Below LOD–114,000 ng/L (influent)	Household wastewater, industrial discharge	[32]
		Below LOD–59,900 ng/L (effluent)		
	Diclofenac	2210–25,000 ng/L (influent)	Household wastewater, industrial discharge	
		360–5000 ng/L (effluent)		
Hormones	Estradiol, progesterone	Typically detected at ng/L levels	Human and veterinary drugs	[32]

**Table 2 ijerph-22-00799-t002:** Common microplastics found in aquatic environments.

Type of Microplastic	Examples	Particle Size Range (μm–mm)	Source	Reference
Fragments	Polystyrene, polyethylene	Oct-00	Plastic waste degradation	[33,34]
Fibers	Polyester, nylon	10–1000	Textile fibers, wastewater	[35,36]
Beads	Polypropylene, acrylic	1–500	Personal care products	[37]

**Table 3 ijerph-22-00799-t003:** Summary of bioaccumulation of pharmaceuticals and microplastics in various aquatic species: contaminant levels and affected organisms.

Contaminant	Species	Bioaccumulation Level	Reference
Pharmaceuticals			
Diphenhydramine	*Perca fluviatilis* (European Perch)	Detected	[137]
Oxazepam	*Perca fluviatilis* (European Perch)	Detected	
Trimethoprim	*Perca fluviatilis* (European Perch)	Not detected	
Diclofenac	*Perca fluviatilis* (European Perch)	Not detected	
Hydroxyzine	*Perca fluviatilis* (European Perch)	Detected	
Microplastics			
Various microplastics	Oncorhynchus tshawytscha (Chinook Salmon)	1.15 particles/individual	[138,139,140]
Various microplastics	Siganus luridus (Dusky Spinefoot)	3.13 particles/individual	
Various microplastics	Liza aurata (Golden Grey Mullet)	3.26 particles/individual	
Various microplastics	Mullus barbatus (Red Mullet)	1.39 particles/individual	
Various microplastics	Sardina pilchardus (European Pilchard)	2.14 particles/individual	
Various microplastics	Scomber japonicus (Atlantic Chub Mackerel)	6.71 particles/individual	
Various microplastics	Mytilus edulis (Blue Mussel)	1.23 particles/individual	
Various microplastics	Copepoda spp. (Copepods)	0.33 particles/individual	
Various microplastics	Cerastoderma edule (Common Cockle)	4.30 particles/individual	
Various microplastics	Hediste diversicolor (Ragworm)	2.70 particles/individual	
Various microplastics	Pelecyora isocardia (Bivalve Mollusk)	1.50 particles/individual	
Various microplastics	Scolelepis squamata (Polychaete Worm)	0.60 particles/individual	
Various microplastics	Scrobicularia plana (Peppery Furrow Shell)	3.30 particles/individual	
Various microplastics	Senilia senilis (Bivalve Mollusk)	1.00 particles/individual	
Various microplastics	Diopatra neapolitana (Polychaete Worm)	1.00 particles/individual	
Various microplastics	Glycera alba (Polychaete Worm)	3.00 particles/individual	

## Data Availability

This manuscript is a review article and does not contain any primary data.

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
