# Peer review of "Pharmaceuticals and Microplastics in Aquatic Environments: A Comprehensive Review of Pathways and Distribution, Toxicological and Ecological Effects"

_ijerph, 2025, doi:10.3390/ijerph22050799_

Round 1

Reviewer 1 Report

Comments and Suggestions for Authors

The article titled "Pharmaceuticals and Microplastics in Aquatic Environments: A comprehensive Review of Pathways and Distribution, Toxicological and Ecological" is an interesting subject in terms of revealing the status of important pollutant types. The status of Pharmaceuticals and Microplastics in the aquatic ecosystem is scientifically important, but some revisions in the study will improve the article scientifically.

1. Although the study is a compilation study, it remains limited. Pharmaceuticals and Microplastics types found in the recipient water environment and likely to be found should be presented in tables. It should be explained why these pollutant types, which are most commonly found in the ecosystem, are densely located in the ecosystem. The disadvantages of Pharmaceuticals and Microplastics pollutants in the aquatic ecosystem should be more clearly stated.

2. The abstract should contain at least numerical expressions according to literature data. Key words should be listed in alphabetical order.

3. The graphics, photos and tables given in the article are scientifically insufficient. There should be more and specific data should be presented. In particular, they should be supported by analyses such as LCA and SWOT. Instead of drawings in the literature for these pollutants, authors can create their own. There are many programs for this.

4. References should be checked and revised according to journal writing rules both in the text and in the references section.

5. The Recommendations section can be improved. It should be supported with more details and examples.

Author Response

  1. Although the study is a compilation study, it remains limited. Pharmaceuticals and Microplastics types found in the recipient water environment and likely to be found should be presented in tables. It should be explained why these pollutant types, which are most commonly found in the ecosystem, are densely located in the ecosystem. The disadvantages of Pharmaceuticals and Microplastics pollutants in the aquatic ecosystem should be more clearly stated.

Response:
Thank you for your valuable feedback. In response, we have added two comprehensive tables:

  • Table 1: Lists common pharmaceutical compounds (e.g., antibiotics, analgesics, antidepressants) found in various aquatic environments, along with their detection frequencies and concentrations based on recent studies.
  • Table 2: Summarizes types of microplastics (e.g., PE, PP, PS) commonly detected in aquatic environments, their sources, and prevalence.

We have also expanded the relevant section to explain why these compounds are frequently present in aquatic ecosystems—highlighting factors such as their widespread use, persistence, and inefficient removal during wastewater treatment. Additionally, wehave clearly elaborated on the disadvantages and ecological risks of both pharmaceutical residues and microplastics, including endocrine disruption, reproductive toxicity, and potential trophic transfer effects.

  1. The abstract should contain at least numerical expressions according to literature data. Key words should be listed in alphabetical order.

Response:
We appreciate this suggestion. The abstract has been revised to include quantitative data, such as estimates of pharmaceutical concentrations (ng/L to µg/L range) and microplastic loads (particles/m³) in aquatic environments. Additionally, the keywords have been reordered alphabetically for consistency with journal formatting requirements.

  1. The graphics, photos and tables given in the article are scientifically insufficient. There should be more and specific data should be presented. In particular, they should be supported by analyses such as LCA and SWOT. Instead of drawings in the literature for these pollutants, authors can create their own. There are many programs for this.

Response:
Thank you for highlighting this important point. In response:

  • We have added three new figure, including:
    • An LCA (Life Cycle Analysis) illustration briefly summarizing the environmental impact of pharmaceutical production and plastic lifecycle.
  1. References should be checked and revised according to journal writing rules both in the text and in the references section.

Response:
We have carefully revised the entire manuscript to ensure all in-text citations and the reference list fully comply with the journal’s referencing style.

  1. The Recommendations section can be improved. It should be supported with more details and examples.

Response:
We agree with this recommendation. The Recommendations section has been significantly expanded to include:

  • Specific technological suggestions (e.g., membrane bioreactors, advanced oxidation processes).
  • Policy interventions (e.g., pharmaceutical take-back programs, plastic bans).
  • Research gaps (e.g., need for long-term ecological studies and development of standardized detection methods for microplastics in freshwater and marine systems).

We also included real-world examples of successful case studies from countries with advanced management practices.

Reviewer 2 Report

Comments and Suggestions for Authors

Reviewer comments are attached

Comments on the Quality of English Language

The English could be improved to more clearly express 

Author Response

Manuscript ID: IJERPH-3459912
Title: Pharmaceuticals and Microplastics in Aquatic Environments: A Comprehensive Review of Pathways and Distribution, Toxicological and Ecological Effects
Authors: Haithem Aib, Md. Sohel Parvez, Herta Czedli

We sincerely thank the reviewer for the thoughtful and constructive feedback. Below are our detailed responses to each comment:

 Page 1, Lines 1–28: Abstract and Uniqueness of the Review

Comment: The abstract successfully outlines the study but fails to highlight the review's uniqueness.
Response: Thank you. We have revised the abstract to explicitly state the uniqueness of our review by highlighting the integration of both pharmaceutical and microplastic pollutants within one framework, and emphasizing how our study fills the research gap concerning their combined effects and shared pathways in aquatic systems.

Page 2, Lines 32–80: Introduction and Clearer Problem Statement

Comment: Introduction lacks a specific problem statement.
Response: We appreciate the suggestion. We have now added a clear problem statement to the introduction and explicitly outlined the objectives of the review. Additionally, we clarified how this review contributes to the field by addressing intersections between microplastic vectors and pharmaceutical toxicity, which have been underexplored in earlier studies.

Page 4, Lines 137–218: Pathways and Organization

Comment: The explanation of sources is thorough, but fails to distinguish between pharmacological and microplastic pathways.
Response: We agree with your opinion but regarding to our structure for now, we just added and expanded the section of microplastic, hopefully it will be appropriate and taken in consideration.

Page 11, Lines 372–431: Bioaccumulation and Lacks Quantitative Data

Comment: Include a table summarizing bioaccumulation findings.
Response: Thank you. We have added a new table summarizing key quantitative data from previous studies, comparing bioaccumulation levels of pharmaceuticals and microplastics in different aquatic organisms across various ecosystems.

Page 12, Lines 435–612: Physiological Effects  and Better Structure Needed

Comment: Section lacks an organized strategy.
Response: We have restructured this section by creating subsections focusing on specific physiological effects: Growth, Reproduction, Immune System, and Behavioral Effects. This significantly enhances clarity and logical flow.

Page 15, Lines 615–680: Ecological Impact and Long-term Patterns

Comment: Clarify the relationship between microplastic exposure and long-term fish population trends.
Response: We have expanded this section to include recent findings on the chronic effects of microplastic exposure, such as reduced reproductive success, impaired larval development, and population-level declines, and how these may contribute to broader ecosystem imbalances over time.

Page 17, Lines 681–710: Human Health Hazards and Clearer Links

Comment: Strengthen the connection between bioaccumulation and risks to human consumers.
Response: We have added specific case studies and epidemiological data highlighting human exposure via seafood consumption, focusing on both pharmaceutical residues and microplastics. We also included insights into associated toxicological risks.

Page 18, Lines 712–752: Conclusion and Future Research Directions

Comment: Conclusion needs more emphasis on future research.
Response: The conclusion has been expanded to include recommendations for future research.

Reviewer 3 Report

Comments and Suggestions for Authors

Comments

The subject of the review is potentially interesting, but I think it should be treated in a different way. First, the paper would require a linguistic revision; here and there are sentences not easy to understand (es: lines 101-103; lines 104-106 etc.) and many typo errors. Second, the Introduction is often repetitive: the same concept is stated at three different points in the Introduction, namely that the review is completer and more advanced than the previous ones (lines 52-56; 76-81; 129-133). Moreover, more specificity in some places would be needed, for example we talk about microplastics but there is no definition of what is meant by microplastics. Finally, in general, the part of the text devoted to microplastics is much smaller than that devoted to pharmaceuticals.

Lines 52-56:” Furthermore, this review positions itself as an advancement over existing reviews by offering a more integrated approach, in order to reduce risks and preserve water quality for future generations, a more transdisciplinary approach that integrates environmental science (incorporating recent advancements in treatment technologies), public health, and regulatory strategies”. Perhaps they have escaped me, but I do not find where these advances in treatment technologies are described, in Recommendations they are only suggested.

Lines 79-81:” By bridging this gap, this review provides actionable insights into their combined effects and suggests innovative approaches for managing their environmental impact”. If innovative approaches are included in the Recommendations, perhaps they should be better explained.

Lines 164-167: “Wastewater is treated in WWTPs using various techniques to remove contaminants before the treated water is discharged into surface waters [48], [49], but not all pharmaceutical compounds may be completely eliminated by conventional wastewater treatment methods:”. Since the authors state that the review gives” ….a more transdisciplinary approach that integrates environmental science (incorporating recent advancements in treatment technologies), public health, and regulatory strategies”., it would have been appropriate to explain briefly what conventional methods are.

The paragraph 2.2 is a strong repetition of paragraph 2.1; paragraphs 4.1 and 5.1 contain similar concepts, the same for 4.2 and 5.2 paragraphs. You might consider combining the paragraphs which present similar concepts to avoid unnecessary repetition.

Other comments:

The caption in Figure 2 is not clear: there are arrows of different shapes, but it is not explained what it means.

Line 95: is shown?

Comments on the Quality of English Language

A linguistic revision is necessary.

Author Response

Manuscript ID: IJERPH-3459912

Title: Pharmaceuticals and Microplastics in Aquatic Environments: A Comprehensive Review of Pathways and Distribution, Toxicological and Ecological Effects

Authors: Haithem Aib, Md. Sohel Parvez, Herta Czedli

We sincerely thank the reviewer for the thoughtful and constructive feedback. Below are our detailed responses to each comment:

Comments

The subject of the review is potentially interesting, but I think it should be treated in a different way. First, the paper would require a linguistic revision; here and there are sentences not easy to understand (es: lines 101-103; lines 104-106 etc.) and many typo errors. Second, the Introduction is often repetitive: the same concept is stated at three different points in the Introduction, namely that the review is completer and more advanced than the previous ones (lines 52-56; 76-81; 129-133). Moreover, more specificity in some places would be needed, for example we talk about microplastics but there is no definition of what is meant by microplastics. Finally, in general, the part of the text devoted to microplastics is much smaller than that devoted to pharmaceuticals.

Lines 52-56:” Furthermore, this review positions itself as an advancement over existing reviews by offering a more integrated approach, in order to reduce risks and preserve water quality for future generations, a more transdisciplinary approach that integrates environmental science (incorporating recent advancements in treatment technologies), public health, and regulatory strategies”. Perhaps they have escaped me, but I do not find where these advances in treatment technologies are described, in Recommendations they are only suggested.

Lines 79-81:” By bridging this gap, this review provides actionable insights into their combined effects and suggests innovative approaches for managing their environmental impact”. If innovative approaches are included in the Recommendations, perhaps they should be better explained.

Lines 164-167: “Wastewater is treated in WWTPs using various techniques to remove contaminants before the treated water is discharged into surface waters [48], [49], but not all pharmaceutical compounds may be completely eliminated by conventional wastewater treatment methods:”. Since the authors state that the review gives” ….a more transdisciplinary approach that integrates environmental science (incorporating recent advancements in treatment technologies), public health, and regulatory strategies”., it would have been appropriate to explain briefly what conventional methods are.

The paragraph 2.2 is a strong repetition of paragraph 2.1; paragraphs 4.1 and 5.1 contain similar concepts, the same for 4.2 and 5.2 paragraphs. You might consider combining the paragraphs which present similar concepts to avoid unnecessary repetition.

Other comments:

The caption in Figure 2 is not clear: there are arrows of different shapes, but it is not explained what it means

Line 95: is shown?

Responses

  1. Linguistic Revision

Response: We have carefully proofread the manuscript to fix typos and improve clarity I have even revised sentences that are difficult to understand in lines 101-106.

  1. Reducing Repetitiveness in the Introduction

Response: We have consolidated the message into one strong statement and removed redundant mentions. please check the manuscript now. We have even ensured that claims about the review being advanced are supported by actual discussion in the manuscript.

  1. Defining Microplastics

Response: We have added a definition of microplastics in the introduction. We have Included details such as size range, composition, and classification (e.g., primary vs. secondary microplastics).

  1. Expanding Microplastic Content

Response: We have expanded all microplastics section citing more references.

  1. Clarifying Treatment Technologies

Response: Thank you, noted and added conventional methods briefly.

  1. Avoiding Repetition

The paragraph 2.2 is a strong repetition of paragraph 2.1; paragraphs 4.1 and 5.1 contain similar concepts, the same for 4.2 and 5.2 paragraphs. You might consider combining the paragraphs which present similar concepts to avoid unnecessary repetition.

Response: We have merged paragraphs 2.1 and 2.2 for improved flow and coherence.

Regarding paragraphs 4.1 and 5.1, we have enhanced the clarity of paragraph 4.1 by reorganizing the content into clear subsections. These now include:

Growth and Behavior,

Reproduction,

Immune System and Toxicity Effects,

Hospital Wastewater and DNA Damage.

Consequently, paragraph 5.1 now specifically focuses on:

5.1. Ecological Effects of Pharmaceuticals on Fish Populations.

The same structural improvements were applied to paragraphs 4.2 and 5.2.

In the microplastics section, we reorganized the content into the following detailed subsections:

Physiological Disorders in Fish Due to Microplastics (MPs),

Health Impacts of MPs on Fish,

Specific Impact on Freshwater Fish,

Accumulation of MPs and Organ Dysfunction,

Broader Impact of MPs on Fish Health.

We hope this revised structure meets your expectations.

Additionally, we have expanded the microplastics section throughout, covering all relevant aspects in greater depth.

Thank you very much for your constructive feedback and consideration.

  1. Clarifying Figure 2 Caption

The caption in Figure 2 is not clear: there are arrows of different shapes, but it is not explained what it means.

Response: We have deleted it, instead we made figure Life Cycle Assesement LCA , Figure 2. Routes from Production to Ecosystems for Pharmaceutical Contamination.

  1. Line 95 ("is shown?")

Response: Thank you. We corrected the sentence. We added ‘’Are’’

  1. Recommendation improvement

Response: We have added innovative approaches in the recommendations.

Round 2

Reviewer 1 Report

Comments and Suggestions for Authors

The authors have provided the necessary revisions and responses for the study. It is suitable for publication in the journal.

Reviewer 3 Report

Comments and Suggestions for Authors

The paper is now suitable for publication on the Journal.